environmental science

specialty coffees, public policy, *sustainabilities* in Brazil

**Author for correspondence:**
Augusto César Pinheiro da Silva
e-mail: augustoc@puc-rio.br

†PhD in Geography (2019) from PGE of PUC-Rio and Professor on CEFET-Maracanã (Rio de Janeiro).
‡PhD in Geography (2005) from PPGG of UFRJ and Associate Professor on PGE of PUC-Rio.

# Political architectures in the municipality of Varre-Sai (Brazil): for *sustainabilities* in 'specialty coffees' production management

Marcelo Orozco Morais[1,†] and

Augusto César Pinheiro da Silva[2,‡]

[1]Pontifical Catholic University of Rio de Janeiro, Rio de Janeiro, RJ, Brazil
[2]Geografia e Meio Ambiente, PUC-Rio, Rio de Janeiro, Rio de Janeiro 22451-900, Brazil

The coffee cultivation has historically had great importance in Rio de Janeiro's social-spatial organization, and it is once again in the spotlight of the state's economic scenario. The novelty is that the local production has been renewed, and, now, the state surpasses the old status of 'low-quality coffee producer' and achieves the 'specialty coffee' market which, in addition to quality, also values environmental, social and economic *sustainabilities*[1]. This change has caused a series of transformations in the social-spatial and economic realities of some municipalities in the state of Rio de Janeiro. This article analyses the possible *sustainabilities* existing in the production process of this new coffee profile, based on the reality of the municipality of Varre-Sai, located in the northwest of Rio de Janeiro state. The objective is to find out the main actors and to elucidate the political network involved in this process.

## 1. Presentation

After a very long crisis that took more than 100 years, since the end of the nineteenth century, the coffee farming in Rio de Janeiro's state almost disappeared in the 1960s, and it 'returned' to be produced in the state in the 1990s under new productive bases focused on mass production. This period coincided with the realization of the first quality coffee contest in the country, won by the producers of Patrocínio (MG), located in the

[1]The word 'sustainability' is commonly purposefully pluralized in Brazilian Social Sciences when the authors have the intention of aggregating multiple forms of sustainability, and not only one specific form, which is the most common interpretation when the word is in its singular form.

**Figure 1.** Location of coffee producing farms in Varre-Sai Municipality (Brazil).

'Triângulo Mineiro' region [1]. Beyond the edaphoclimatic characteristics that favour coffee production, that region has been benefited with the establishment of a strong political articulation between its producers and the state and federal governments, through which the country's first coffee *terroir*[2], the 'Cerrado Coffee', was delimited [1]. It was one of the first steps towards the explosion of specialty coffee production in Brazil.

This process of certificating the origin and quality of coffee crops was accompanied by the growth of the specialty coffee consumer market in Brazil and worldwide. Nonetheless, it is important to highlight that this market niche does not only value flavour, but also, and increasingly, the environmental, social and economic *sustainabilities* involved in the process of production.

From the mid-twentieth century onwards, the reassessment of the notion of development was progressively intensified towards the concept of sustainable development, in a way that the understanding of the sustainable development discussion would progressively run through changes in the development paradigm. However, as the idea of sustainable development lacks a strict definition, the 'sustainable' actions were conditioned by what each proponent understood by sustainability and sustainable development and, consequently, by the dimensions that each definition privileges.

Emphasizing the environmental, social and economic dimensions of sustainability, we present a case study related to the municipality of Varre-Sai in Rio de Janeiro (Brazil), which since the beginning of 2010s has stood out on specialty coffees production. Until then, this locality was acknowledged by its traditional coffee production, for which it has been called 'the Capital of Coffee'. Since then Varre-Sai has transformed its productive, socio-economic and environmental engagement as it relocates itself in the coffee market.

This article is structured in four parts as follows. Section 2 discusses the recent transformations in the cultivation of coffee in the state of Rio de Janeiro, emphasizing its institutional consequences, i.e. it describes how the changes verified on the national and international market and on the political arrangements in the Brazilian coffee cultivation led the construction of a framework of territorial public policies essential for redirecting the coffee production towards a gain in quality and value. Section 3 briefly presents the Varre-Sai municipality, in order to describe its socio-spatial formation and highlight the network of institutional and non-institutional actors tangled in specialty coffees production, and also verify the main public policies created to make it possible for Varre-Sai's producers to enter this new market. Section 4 connects the results of §§2 and 3 with the debate concerning sustainability, i.e. after a brief theoretical conceptualization of sustainability, it presents an analysis of how the social, spatial and environmental dimensions of sustainability are exhibited in the coffee production of Varre-Sai, also introducing its challenges. Section 5 presents the conclusions figure 1.

---

[2]*Terroir* appoints 'a limited land extent considered from its agricultural aptitudes point of view' [2].

# 2. The New Rural: the case of specialty coffee production

The coffee market have been marked, in the recent years, by the emergency of the so-called 'Third Wave' of coffee—a term signed by the American barista Trish Skeie [3], referring to the increase in the seek for quality in a recent period.[3] This period is characterized by a segmentation in the market in which a change in the consumers' mentality can be identified, which led to a resignification and even a reinvention of the drink. This transformation of the consumers' demand is aligned with the so-called Brazilian New Rural context [5].

The conformation of current Brazilian rural space is more complex than it was in the old times, led by the coffee barons. The so-called 'New Rural', a term signed by Graziano da Silva [5], is configured, beyond modern agriculture and non-agriculture activities, by the valuing of increasingly specialized activities, located in special niche markets. This new reality, according to the author, overthrows some dated ideas concerning the Brazilian rural space, such as the association between rural and backwardness, rural and agriculture production, also as placing the rural exodus as something inevitable.

This does not mean that the 'old rural' has been overcome. In many aspects, some problems ended up been ratified, such as social disparities, and environmental problems. Thereby, the incorporation of the sustainability agenda on agriculture, as one of the characteristics of this process, is related to the emergency of the complex movements and discussions concerning the promotion of sustainable development. Therefore, the specialty coffee production represents an attempt to overcome technical and, consequently, economic, social and environmental problems of the activity through the valorization of quality over quantity, what can be noticed as increasingly bound to the introduction of sustainable practices in the production as a whole.

## 2.1. New bureaucratic structures and the repositioning of public and private agents

State regulation was one of the main factors that led to a deterioration in Brazilian coffees' prestige in the international market. Variations in quality created problems for buyers, since they recurrently had to discard or return the lots. Besides, in the internal market, the coffee sold was basically what was rejected abroad, due to its bad quality. According to Saes & Spers [6], adding corn, barley, rye, caramel, husks, and straw in the bag of coffee was a common habit.[4]

Since the 1990s, new market trends (internal and external) boosted agricultural development towards higher quality coffee production. The political decentralization inaugurated with the 1988 Federal Constitution associated with the adoption of a series of liberalizing measures at the beginning of Fernando Collor's government (1989–1992) extinguished the *Instituto Brasileiro do Café* (IBC)[5], which worked as a real 'umbrella' for coffee producers, and established the bases for the diffusion of the specialty coffees in the country. This new approach followed the new international market demands, counting increasingly on a new organization of production based on the partnership between private initiative and the State, and on the creation of new mechanisms in which quality and *sustainabilities* are valorized.

## 2.2. Territorial public policies in Brazil

The concept of territorial development has increasingly been mobilized as 'one of the methods for considering the ways in which State and local actors act to promote development policies' [7, p. 23]. The policies of territorial development emphasize the importance of local actors in the construction of positive synergies in their territories in articulation with the public sphere. In Brazil, the national government has led this kind of policies. Thus, in each delimited area, councils were established, and they were constituted by local community members and representatives of public power. These

---

[3]In the past two years, due to the movement acceleration on the globalized world, specialists in the sector increasingly signal the 'arrival' of a Fourth Wave that challenges the current system through new advances and innovations on this market [4, p. 75]. The multiplication of technological innovations that approach the final consumer stands out, in a way that also the market giants have been progressively entering into this new niche market.

[4]According to the authors, this practice was illegal, since the 12/1978 Resolution, of the *Comissão Nacional de Normas e Padrões Alimentares* (CNNPA)—in English, Food Norms and Standards National Commission—prohibited the addition of strange products in coffee, admitting a maximum tolerance of 1% of impurities (husks and sticks) and Resolution 2 of 1990 of IBC prohibited industrialization and commercialization of coffee that was altered or adulterated in any way [6, p. 356].

[5]In English, Brazilian Coffee Institute.

councils focused attention on the local realities, and as a result, they produced a demand for specific projects addressing their localities.

The perspective of territorial development planning dedicated to coffee cultivation commenced in Brazil in the 1990s. Since then, a series of public policies started to be generated from this theoretical-methodological framework. Among the specific territorial policies applied to the rural, we highlight, on the federal scale, the creation of the *Programa Nacional de Fortalecimento da Agricultura Familiar* (PRONAF)[6] and the Ministry of Agrarian Development. Within Rio de Janeiro State, we highlight the *Programa de Desenvolvimento Rural Sustentável em Microbacias Hidrográficas do Estado do Rio de Janeiro* (Rio Rural)[7] and rural extensionism.

### 2.2.1. PRONAF's policies

PRONAF was created in 1995, at a time when the cost and shortage of credit were the main concerns of farmers [8]. Those conditions were a result of the conservative modernization of agriculture led by the military regime [9], which gave priority to large and medium patron-producers seeking to increase exportation, in prejudice of small and/or family agriculture. Thereby, the political articulation of small/family/low-income producers was essential to the formulation and implantation of specific rural development policies [10, p. 58].

According to the Ministry of Agrarian Development, PRONAF is characterized by the small farmers' participation since its creation. As a result, its actions are mainly based in the bottom-up model, involving organizations of farmers mobilized in constant dialogue with public policy managers. Thus, operationally, the PRONAF is focused on four great lines of action: (i) financing of production; (ii) financing of municipal infrastructure and services; (iii) family farmers training and professionalization; and (iv) financing of rural extension and research.

The definition of which groups would be contemplated as beneficiaries have changed as the definition of family farmers changed. Thereby, according to Panzutti & Monteiro [11, p. 135], currently, there are 15 groups established with the following requirements: (i) explore a portion of land in the condition of owners, settlers, squatters, tenants, partners or sharecroppers that use predominantly family labour and have up to two permanent employees; (ii) do not hold any title or areas superior to four fiscal modules; (iii) have a gross family income with at least 80% originating from agricultural and non-agricultural activities developed in the establishment; and (iv) reside on the property or at a near place [11].

Currently, PRONAF's resources come from the following sources: Special Deposits of *Fundo de Amparo ao Trabalhador*[8], bank liabilities, constitutional funds (FNE, FCO and FNO), rural savings, *Orçamento Geral da União* (OGU)[9], equalizable own resources and *Fundo de Defesa da Economia Cafeeira* (Funcafe)[10].

The Funcafe operates with financing lines to support the coffee sector and destines a small percentage of its budget to family agriculture's coffee-growing financing. This dynamic is essential for the expansion of the activity in the country, strengthening the bases for the diversification of production in search for the improvement of its quality.

### 2.2.2. Rio Rural policies

Rio Rural has as its primary objectives the increase in producers' income and, at the same time, the conservation of natural resources, improving the quality of life of the communities involved in the project. Therefore, its strategy is directed to promoting participatory planning in the micro-watersheds in order to highlight local, economic, environmental and social specificities, peculiar to each micro-watershed, in the process of discussion and decision.

An important characteristic of the Rio Rural is the direct involvement of the community in its actions. The dwellers are invited to collect data themselves, and they are encouraged to develop suggestions that could lead to the solution of the diagnosed problems. In this way, they create adaptations to their own realities of sustainable management practices of natural resources on rural properties, through participatory researches under the responsibility of the *Empresa de Pesquisa*

[6]In English, Family Agriculture Strengthening National Programme.

[7]In English, Programme of Sustainable Rural Development in Micro Watershed of the State of Rio de Janeiro.

[8]In English, Worker's Assistance Found.

[9]In English, Union's General Budget.

[10]In English, Coffee Economy Defence Fund.

*Agropecuária do Estado do Rio de Janeiro* (Pesagro-Rio)[11]. According to Pesagro-Rio, these researches have the objective of adjusting sustainable technologies to the specific social, economic and environmental conditions of each micro-watershed and to each producer participating in the process.

### 2.2.3. Rural extension policy

The access to credit relies on agencies and companies of the *Agência Técnica de Extensão Rural* (ATER)[12], responsible for the diffusion of technical knowledge that leads to an increase in productivity and also in the quality and sustainability of the production. This guarantees greater competitiveness in the market to the small producers [12]. In Rio de Janeiro, for example, technicians of the Emater-Rio are responsible for promoting sustainable and agroecological agriculture in conjunction with Rio Rural, the most important programme of the state in serving Rio de Janeiro's family production.

The role of rural extension on specialty coffees production is to call for an increase in the articulation of stages upstream and downstream of the production chain. In other words, it requires attention to coffee processing and commercialization phases. This is made in order to raise value of the whole coffee chain. Traditionally, as coffee is sold green to the market for specialty coffees sellers, the sellers' brand, and not the producer's ones, are strengthened, which decreases market strength for the producers. Therewith, ATER institutions active in the Brazilian coffee cultivation are increasingly working towards the correction of market imperfections, as it will be seen in Varre-Sai's case. They comprehended that without a favourable environment for the small producer in the outer markets, there is the risk of failure in the production of quality special coffees, and in extension to the rural extension policy.

# 3. Contextualizing the municipality of Varre-Sai

The northwest region of Rio de Janeiro was one of the last expansion zones where coffee culture succeeded, especially between the end of the nineteenth century and throughout the twentieth century. In the first decades of the twentieth century, most of the Brazilian coffee exports [13] had departed from that region, which became the largest national producer at that time, and is still the largest producer of the state.

Varre-Sai's history is marked by a succession of territorial separations and annexations, which makes it difficult to have precise access to specific data on coffee production related to the exact portion of territory that the municipality is currently inserted in. In addition, its actual perimeter belonged, until 1947 to the municipality of Itaperuna, and from 1947 to January 1991, to the municipality of Natividade do Carangola, and so, it only appears in the Agriculture Censuses after 1991, when it was emancipated.

The territory that comprehends Varre-Sai is located on an old muleteer's troop route. The toponymy Varre-Sai is said to have come from an old ranch where the troopers used to stay. There are disagreements concerning this explanation; however, the influence of the passage of the muleteer troop on this region's colonization is undeniable:

> Due to its location on the extreme northwest of Rio de Janeiro, on the border with the state of Espírito Santo, at an altitude of 682 meters, the muleteer troop were, if not the only one, one of the few means of communication between Varre-Sai and the rest of the Province. They carried goods, news, packages, and letters. [14, p. 492]

From the eighteenth century onwards, the northwest of Rio de Janeiro was progressively occupied by immigrants from Rio de Janeiro, Espírito Santo and, mainly, Minas Gerais. These first occupants performed a fragmented occupation of the region that today comprehends the Itaperuna, Natividade, Porciúncula and Varre-Sai municipalities. Its agricultural production was based both on family and slave labour. Thereby, in contrast with what happened in the Valley of Paraíba River, a region known for its coffee production, the northwest region did not produce an agrarian aristocracy; instead, its land structure was characterized by the presence of small and medium properties that prevails until today [15,16].

The region experienced significant incoming of European immigrants by the end of the nineteenth century, most of them were Italians. These immigrants were, in the majority, farmers and smallholders—mostly illiterate—that lost their land to the government. Nevertheless, rapidly, these immigrant settlers paid their debts and became small land or real estate owners. This social mobility is attributed to the prosperity in coffee production, to which activity most of these immigrants have dedicated their work [16].

---

[11]In English, Agriculture Research Company of the State of Rio de Janeiro.

[12]In English, Rural Extension Technical Agency.

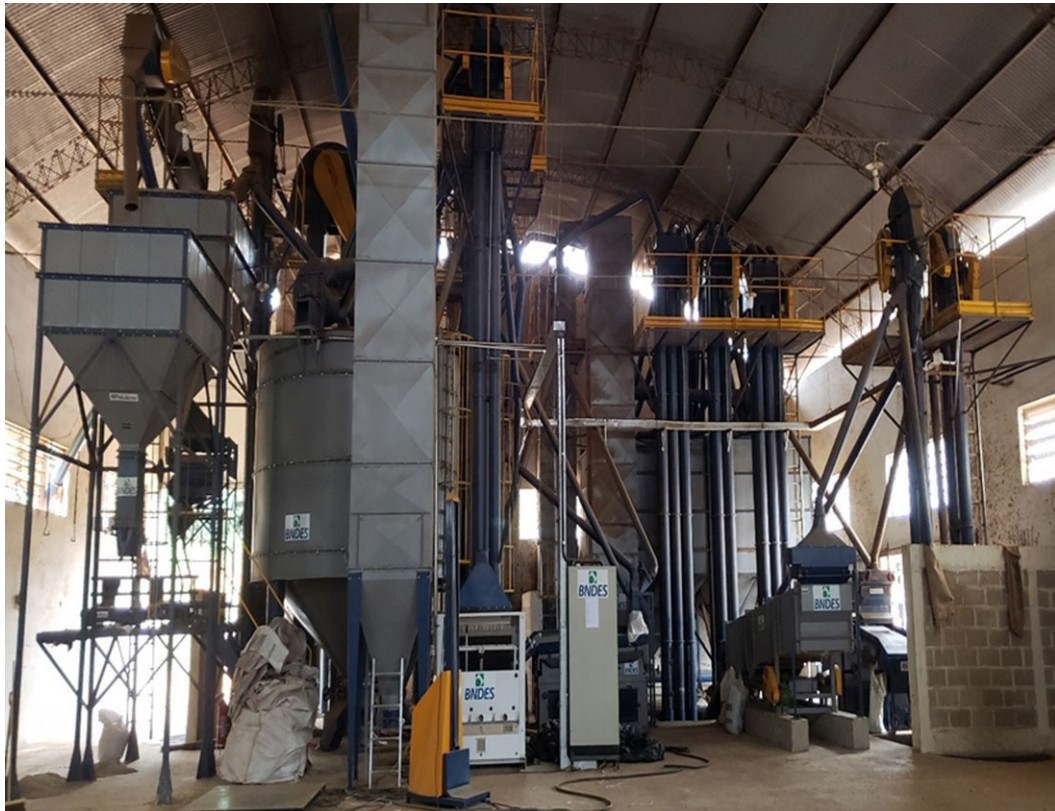

**Figure 2.** Coffee processing machinery, COOPERCANOL. Source: author's files. May 2019.

Bartholazzi [17] adds that the Italian immigrants adopted two basic strategies in the host country: (i) they comprehended the potentiality and 'the specificity of the region, the northwest of Rio de Janeiro, which was characterized by being a coffee-growing expansion area, in the republican period, marked by small and mid-property' and (ii) they built solidarity networks [17, p. 11]. According to the author, this last one was essential for 'guaranteeing the possession and permanence of the property, as a representation of safety and protection among the "foreigners" [17, p. 9]. These 'networks' consisted basically of easiness in transactions between compatriots and the advantages of the spatial proximity to their compatriots when making their investments.

Therefore, the land structure and a certain degree of social capital favoured coffee-growing development. The real constraint was the financial limitation of the local producers, i.e. although they own theirs lands, they have reduced autonomy in conducting the productive process, and they also have limitations regarding the possibility of investing in their lands [18]. Furthermore, the productive model historically adopted is characterized by intense land use, causing damage to the environment.

Currently, the northwest region of Rio de Janeiro concentrates 70% of the state's coffee crops, 24% of which are found in Varre-Sai [19]. In 2014, the city was declared the 'Coffee Capital' by the state Law No. 6726. It was due to its expressive results that made it emerge as the largest coffee producer of the state. The municipality has 20% of its area, i.e. 4660 hectares, composed of coffee-planted area (CONAB). Coffee cultivation is one of the municipality's main economic activities, driving about 46 million reais into its economy. Varre-Sai's production is equivalent to approximately 40% of all coffee produced in the Northeast region of Brazil [20] (figure 2).

Varre-Sai's entrance into the specialty coffee market has reframed its participation in the national coffee-growing scenario and has transformed the city's socio-economic reality. We verified that the recent constitution of a political architecture that conjugates public and private actors with a public policy network dedicated to encouraging the production of specialty coffee has made the municipality reach increasingly satisfactory results. Through the reactivation of the *Cooperativa de Café do Noroeste Fluminense* (Coopercanol)[13] in 2014, the extensionists from Emater and Sebrae's agents, along with a group of approximately 20 producers conjugated with the Municipal Secretariat of Agriculture, gave

[13]In English, Rio de Janeiro's Northwest Coffee Cooperation

rise to a series of enterprises. Thus, through Emater, a group of 5313 producers had access to credit programmes, such as PRONAF, totalizing an investment of R$ 44 225 045 in the municipality.

It was possible to verify that there is, in Varre-Sai, a certain degree of trust, especially among the owners, farm workers and the associated institutions (Coopercanol, Sebrae, Emater and the Municipal Secretariat of Agriculture). This cooperative tradition created a favourable environment which, with the aid of Sebrae and Emater, and the availability of public funds, made possible the design of a new political architecture of networks focused on the project of turning Varre-Sai into a reference in the sustainable production of specialty coffees.

The first major initiatives in this way were the courses taken by Professor Flávio Borém from the Federal University of Lavras, which, from 2014 to 2015, brought together a class of coffee growers from Varre-Sai, intending to teach them how to produce specialty coffees, and encouraging them to realize the Specialty Coffee Contest of Varre-Sai, that took place at Coopercanol's headquarters. Such efforts were accomplished by the collaboration of state and non-state actors, which demonstrates the existence of common interest.

Thereby, Varre-Sai's producers managed to reach prominent positions in the specialty coffees scenario in a short period, as could be certified in the Quality Coffee Contest of the State of Rio de Janeiro, held in October 2018, in which producers from that region occupied six places among the 10 finalists. Another concrete example of this entry in specialty coffees world was the participation of Rio de Janeiro in the international coffee fair, held in Belo Horizonte, in 2018. The setting up of a stand with coffees from Rio de Janeiro caused great admiration, since this would not be imaginable some time ago.

# 4. The *sustainabilities* of specialty coffees production in Varre-Sai

Brazil, which has always been very competitive in the coffee market due to the huge volume of production [21], has since 2000 achieving notoriety in the special coffee's market due to its coordinated effort in order to gain quality. It is so, because many Brazilian producers have incorporated a series of practices mobilized by both the public power and the private initiative. The use of guaranteed provenance seedlings, the business management of agricultural property, the participation in quality contests, investments in technological innovation and, not least important, investments in marketing and on the identity of the region has been, according with IBGE [22], some of the practices that characterize this new phase of Brazilian coffee production.

Following this logic, Zambolim [23] proposes a model of representation of the necessary attributes for coffee farming with sustainable quality, in which it would be articulated: (i) environmentally correct production, (ii) certification of origin, (iii) product differentiated by quality, and (iv) social responsibility.

Based on these propositions, we present the possible scenarios of the *sustainabilities* of specialty coffees production in Varre-Sai.

## 4.1. Environmental sustainability

The discussion regarding environmental sustainability is related to a proposal of 'a model of production and consumption compatible with the material ground in which the economy is based, taken as a subsystem of the natural environment' [24, p. 55]. Therefore, it is characterized by respect for the environment and the comprehension of its dynamics. According to Veiga [25], sustainable agriculture production is carried out on three fronts: the combat of agroecosystem's degradation caused by the modernizing process of the twentieth century; the demand for new disciplinary rules for food-based agriculture system and the demand for more appropriate practices for the preservation of natural resources and the supply of healthier food.

According to Dallabrida [26], the shift towards critical environmental awareness requires, among other things, overcoming the reductionist view that summarizes nature as a resource, giving more credit to a systemic view of nature. This shift in 'awareness' has expanded in the municipality and can be perceived in interviews, as in the following excerpt:

> We think about the sustainability issue. (…) And environmental sustainability, which consists of working the right amount of pesticides, always respecting the dosages, with the assistance of an agronomist. Everything within the standards. We reduced drastically the use of pesticides. The persistent control of erosion and soil conservation. So that we always think about producing coffee for 500 years. It is not producing for 5 years and then the soil is over. We think about producing coffee forever. (…) (VARGAS, Sergio. Café Vargas, 13/11/2018)

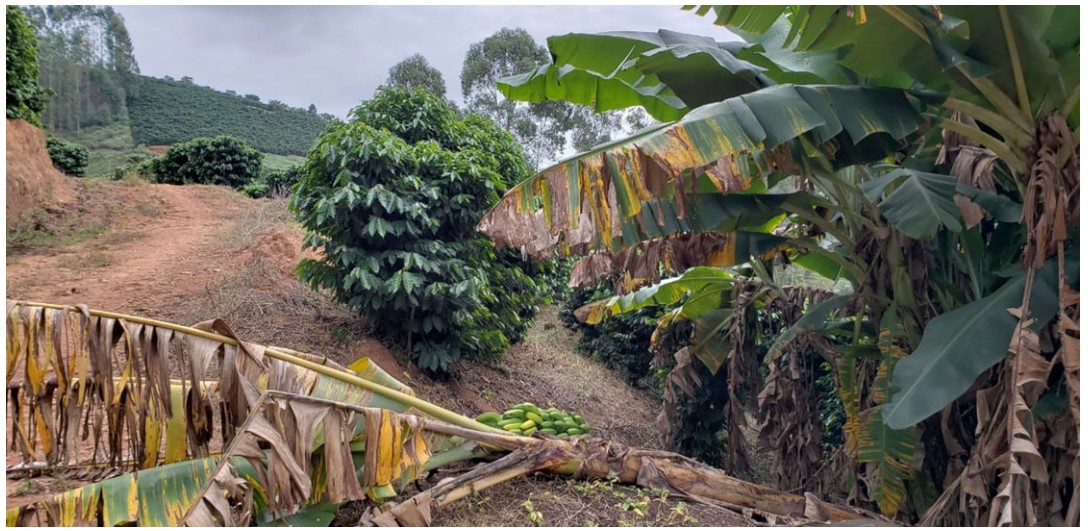

**Figure 3.** Coffee and banana intercrop plantation in Vai e Volta site, Varre-Sai. This conjugation guarantees shadow for the coffee plantation, reduction of erosion and soil fertilization. Source: author's files. May 2019.

The beneficiaries of federal and state financing programmes, primarily PRONAF and Rio Rural, were encouraged to suit their properties with environmentally sustainable practices, such as favouring rainwater infiltration, recovering and/or increasing aquifers' flow and promoting soil conservation. With financial and institutional support, they improved their crops and the specific structures of their properties, by adopting new agroecological technical procedures and the diversification of agricultural activities. As a result, there has been a sensible reduction of erosion processes due to intercropping coffee with other crops (figure 3). Another measure implemented was the implantation of the so-called micro terraces of about 30 cm between the coffee rows in order to facilitate the movement between the rows of the plantation during the harvest and in treatment of plants, as well as making the harvest feasible, because a great part of the coffee plantations is settled in high declivity areas (figures 3 and 4).

## 4.2. Social sustainability

According to Folladori [27], we must put aside the theoretical and schematic field and see how things work in practice, so that we can comprehend what sustainable development and its *sustainabilities* really mean. In his view, environmental sustainability, despite being difficult to measure, is the one that generates the least amount of disagreements about a possible definition. However, social sustainability is the most controversial and ambiguous one [27].

Therefore, the evolution of the debate led to the incorporation of the issue of social participation as an instrument of strengthening social agents to seek social sustainability. In this sense, it is important to highlight the Rio Rural programme, applied in Varre-Sai, which seeks sustainable alternatives for development through participatory practices and methodologies that mobilize farmers and the public, from which they build proposals of intervention, aligned with local interests and needs.

There is a concern to promote autonomy to producers, in an effort to create a culture of seeking technical improvement to establish good management, harvesting, processing and commercialization practices. To do so, a series of courses and lectures have been offered through an articulation between Emater, Sebrae, Municipal Secretariat of Agriculture and Coopernacol, in addition to the collaboration of producers themselves and chemical inputs companies, like Bayer, and the Ministry of Agriculture, Livestock and Supply.

As for the question of labour, it was verified that the use of partnership and sharecropping also predominates in specialty coffee crops. Today, the Coffee Productive Chain of Rio de Janeiro's northwest is characterized by involving 3200 rural producers, predominantly family farmers, according to Emater-Rio data. This type of relation of work is historical in the municipality of Varre-Sai, and it has its origins in the process of colonization of the region with the arrival of the Italian labour, as previously seen.

One of the main problems regarding this type of relation of production is, according to Staduto *et al.* [28], the issue of social protection and sharecroppers exploitation by the producer for carrying out

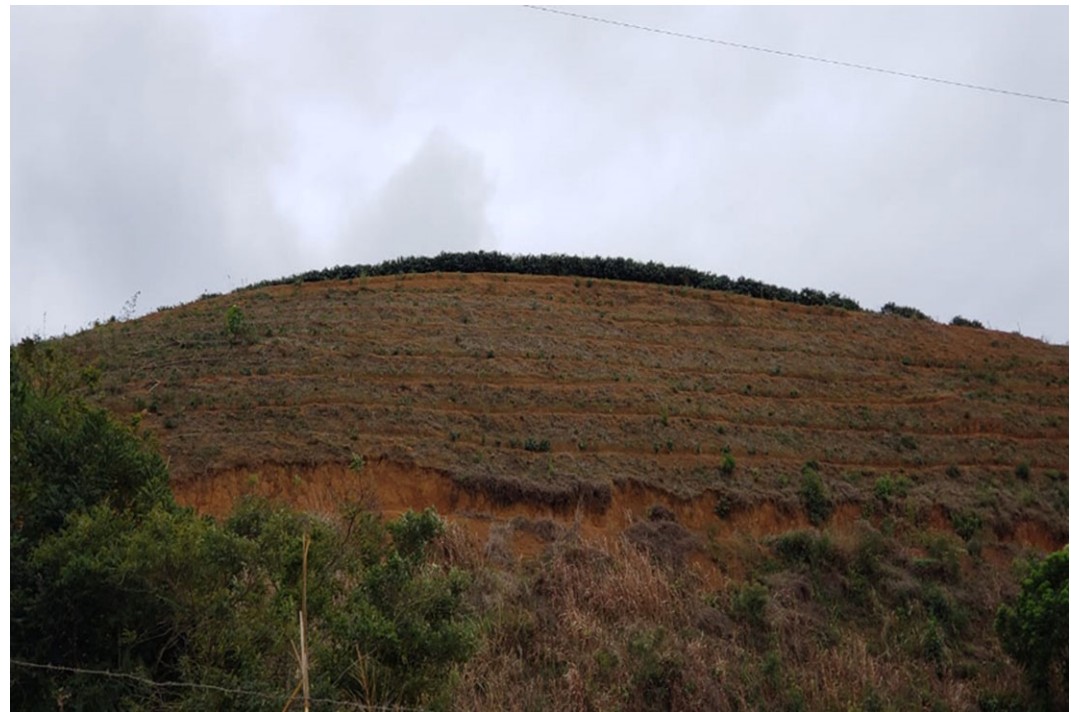

**Figure 4.** Microterraces in a farm: the reduction of erosion and facilitation for harvesting. Source: author's files. August 2019.

activities not provided for in the contract, i.e. the workers with no social guarantees often find themselves submissive to non-contractual orders. The advent of specialty coffees in the municipality has boosted the regularization of work contracts, for it also incurs a greater concern with labour's quality. In addition to the intensification of institutional pressures by the Ministry of Labour, which has required reasonable conditions of work in the crops, the certifiers coveted by specialty coffee's producers, such as Fair Trade, has also demanded certain adjustments regarding the welfare of labour. These pressures have caused some accelerated changes.

Therefore, labour protection has expanded: in case of accidents, these workers are covered by social security, and pregnant women have the right to maternity leave, as well as the elderly to retirement. Besides, the expansion of the activity has incurred in the permanence of the countryside population, reducing rural exodus rates. According to the interviewees, this occurs due to two main factors: (i) improvement of life quality in the countryside, especially for the workers involved in the specialty coffees chain, and (ii) the high costs of living on the urban area (figure 5).

## 4.3. The consequent economic sustainability

Criticism and accountability to private capital as the main responsible for environmental and social degradation process led to a response from these sectors, especially in the innovation field, in a way that the incorporation of sustainable practices (in different dimensions) became a factor of competitiveness and differentiation factor in the market. In the case of specialty coffees:

> The key points that involve the valorization of the product and the elaboration of a quality drink are innumerable, starting at the care in conducting the crop and a well-done post-harvest, to the environmental and social responsibilities that comprehend a production system. However, the characteristics of the productive region, with its singular *terroir* and its intimate relationship with cultivars are of great importance in this process and promote the creation of intense, tasty and even exotic coffees, which are much appreciated by consumers worldwide and reach high values in the market. [29, p. 24]

In effect, public and private political articulations, applied to Varre-Sai, have indicated the 'potential qualities' that are explored on the municipality: the socio-spatial structuration, which was traditionally seen as a constraint, has shown to be more appropriate to the specialty coffee markets. The phenomenon observed in Varre-Sai shows us how the improvement of techniques of production, incorporating new technologies to the crops, has led to an upturn in the municipal (and regional) economy and, as a consequence, has also led to advances in environmental sustainability.

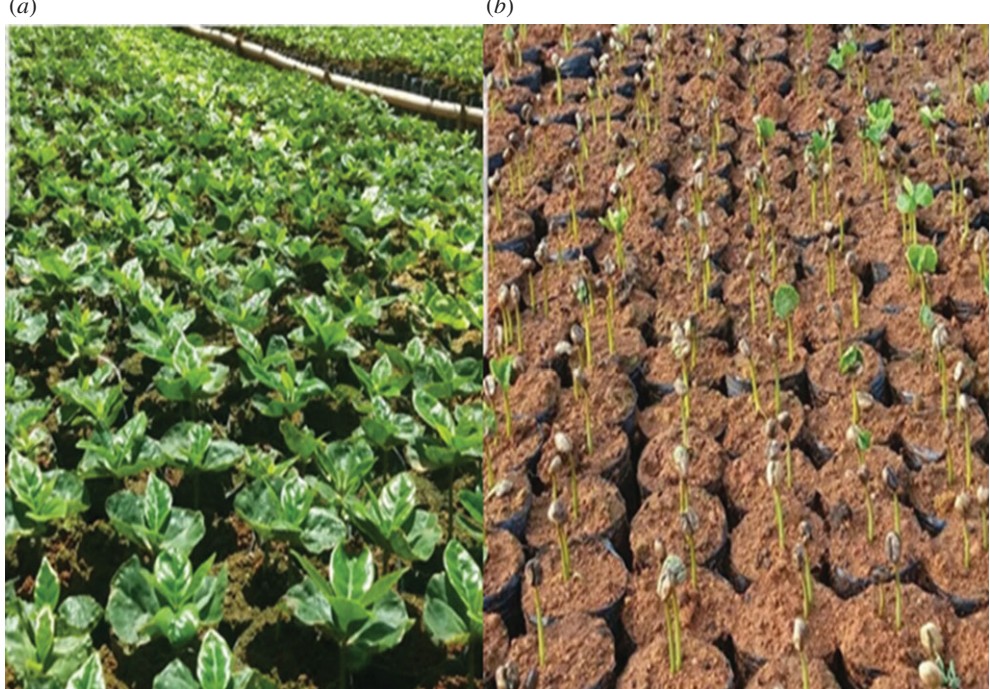

*(a)*     *(b)*

**Figure 5.** Seedling nursery in Varre-Sai. Source: author's files. August 2019.

In Varre-Sai, a group of producers is already able to roast their coffee and develop their own brands, in a process known as 'from seed to cup'. In terms of economic sustainability, this is an important conquest for the family farmers, because they can retain the surplus value of their own product. In the past, farmers would sell their green coffee to a broker, losing the benefit [23]. This practice has been systematically modified, even for those who do not have their own coffee brand or do not roast their own coffee, because it is already possible to classify the coffee produced at Coopernacol's facilities.

It is important to highlight that the specialty coffees chain has also encouraged the diversification of economic activities in the municipality and the expansion of employment (formal and informal). Based on field research and interviews, it was possible to verify this diversification of activities driven by the expansion of specialty coffees production, and manifested in the emergence of commercializing partnerships, as well as the opening of stores dedicated to specialty coffee selling. As an example, we can cite the first coffee shop of Varre-Sai, and the project—still incipient—of creating a touristic circuit.

## 5. Conclusion

Coffee has active participation in the Brazilian economy, with significant participation in the country's exports. However, the traditional model of production adopted in Rio de Janeiro privileged productivity in terms of volume, with no concerns on quality and sustainability, which led to a decline in production and, above all, has made the state become famous for producing coffee of a very low-quality. This fame is hard to remove; however, despite this, strong work has been done by the actors involved in Rio de Janeiro's coffee crops to reverse it.

The entrance of Varre-Sai in the specialty coffees world in a sustainable way is due, largely, to the deregulation that occurred at the beginning of the 1990s, with the extinction of IBC and state regulation of the sector, which allowed an articulation between public and private agents, resulting in a transformation of the productive space of the municipality. This articulation was made possible thanks to the presence of actors that formed a network aiming to improve the quality of the coffee produced locally. As a result, it was proven that this articulation has the potential of expanding specialty coffees production with attention to sustainability, and to create a sustainable high-quality product capable of penetrating in specialized niche markets previously restricted to those regions traditionally involved in the production of specialty coffees, such as the South region of Brazil and the Cerrado of Minas Gerais.

Therefore, a set of public policies that, driven by the social capital present in the municipality and the mutual trust of the actors involved, allowed the overcoming of obstacles, and have culminated in a successful work with a great potential for promoting expansion in specialty coffees production, although surrounded by challenges in multiple scales. The results presented here allow us to affirm that the municipality has great potential that needs to be stimulated with the maintenance and expansion of these public policies. Such policies should stimulate the region's development, while strengthening family farmers, which are, according to Abramovay [30], the most qualified actors for this new scenario that is drawn of a production based on *sustainabilities*.

Data accessibility. This article has no additional data.

Authors' contributions. M.O.M. conducted fieldwork, took photographs, tabulated primary data and conducted interviews; A.C.P.S made technical and academic adjustments. Both authors approved the final version for publication and agreed to be held responsible for the work performed.

Competing interests. We declare we have no competing interests.

Funding. This study was supported by Coordenação de Aperfeiçoamento de Pessoal de Nível Superior (001).

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
