## [Peer Review File · Royal Society Open Science]

Review History

RSOS-201874.R0 (Original submission)

Review form: Reviewer 1 (Clemente Fabregat)

Is the manuscript scientifically sound in its present form?

Yes

Are the interpretations and conclusions justified by the results?

Yes

Is the language acceptable?

Yes

Do you have any ethical concerns with this paper?

No

Have you any concerns about statistical analyses in this paper?

No

Recommendation?

Accept as is

Comments to the Author(s)

It would be advisable that the size and type of the letters of the item: AS SUSTENTABILIDAD DE PRODUÇÃO DE CAFES ESPECIALES EM VARRE-SAI, should be the same of the items: O NOVO RURAL: O CASO DO PRODUÇÃO DE CAFÉS ESPECIAIS, CONTEXTUALIANDO VARRE-SAI, and CONCLUSÕES.

Review form: Reviewer 2**Is the manuscript scientifically sound in its present form?**

Yes

Are the interpretations and conclusions justified by the results?

Yes

Is the language acceptable?

Yes

Do you have any ethical concerns with this paper?

No

Have you any concerns about statistical analyses in this paper?

No

Recommendation?

Accept with minor revision (please list in comments)

Comments to the Author(s)

There are some errors:

page 2, line 47 - that that;

page 3, line 22 - the 'sustainable' actions was;

page 6, lines 17 and 18 - they produced produced;

page 11, line 47 - on the on the;

page 17, line 24 - Thereby, is stimulates.

Some of the literature are not referenced in the references section: Trish Skeie (2002), (CORRÊA, 2009, p.23), (GUANZIROLI, 2007), (DELGADO, 2012), MATTEI, 2010, p. 58), (MARTINO, 2011, p.492), (LAMEGO, 1963), (CONAB), (IBGE, 2017), (IBGE, 2016), (NASCIMENTO, 2012, p.55).

I don't think that the challenges of the coffee production of varre Sai were addressed as stated in page 4, line 5.

It would have been interesting exploring some data from the 2017 agricultural census as well as supplementary data of local institutions on production, land size and work conditions.

Decision letter (RSOS-201874.R0)

Dear Dr Silva

On behalf of the Editors, we are pleased to inform you that your Manuscript RSOS-201874 "POLITICAL ARCHITECTURES IN THE MUNICIPALITY OF VARRE-SAI (Brazil): FOR SUSTAINABILITIES IN "SPECIALTY COFFEES" PRODUCTION MANAGEMENT" has been accepted for publication in Royal Society Open Science subject to minor revision in accordance with the referees' reports. Please find the referees' comments along with any feedback from the Editors below my signature.

Please submit your revised manuscript and required files (see below) no later than 7 days from today's (ie 22nd of February 2021). Note: the ScholarOne system will 'lock' if submission of the revision is attempted 7 or more days after the deadline. If you do not think you will be able to meet this deadline please contact the editorial office immediately.

on behalf of Dr Agnieszka Latawiec (Subject Editor)
openscience@royalsociety.org

Associate Editor Comments to Author (Dr Agnieszka Latawiec):

Dear Authors

The reviewers of your manuscript have returned their comments, which are very positive. I also find your case study very interesting, novel and fit for the Special Collection on Sustainable Land Use. Before the paper is accepted please incorporate the comments of Reviewer 2 and scan your manuscript for typos, I have picked a few.

Congratulations and looking forward to hearing from you.
Please do not hesitate to contact me in case of any questions.

Kind Regards,
Agnieszka Latawiec

Reviewer comments to Author:

Reviewer: 1
Comments to the Author(s)

It would be advisable that the size and type of the letters of the item: AS SUSTENTABILIDAD DE PRODUÇÃO DE CAFES ESPECIALES EM VARRE-SAI, should be the same of the items: O NOVO RURAL: O CASO DO PRODUÇÃO DE CAFÉS ESPECIAIS, CONTEXTUALIANDO VARRE-SAI, and CONCLUSÕES.

Reviewer: 2
Comments to the Author(s)

There are some errors:

page 2, line 47 - that that;
page 3, line 22 - the 'sustainable' actions was;
page 6, lines 17 and 18 - they produced produced;
page 11, line 47 - on the on the;
page 17, line 24 - Thereby, is stimulates.

Some of the literature are not referenced in the references section: Trish Skeie (2002), (CORRÊA, 2009, p.23), (GUANZIROLI, 2007), (DELGADO, 2012), (MATTEI, 2010, p. 58), (MARTINO, 2011, p.492), (LAMEGO, 1963), (CONAB), (IBGE, 2017), (IBGE, 2016), (NASCIMENTO, 2012, p.55).

I don't think that the challenges of the coffee production of varre Sai were addressed as stated in page 4, line 5.

It would have been interesting exploring some data from the 2017 agricultural census as well as supplementary data of local institutions on production, land size and work conditions.

===PREPARING YOUR MANUSCRIPT===

Your revised paper should include the changes requested by the referees and Editors of your manuscript. You should provide two versions of this manuscript and both versions must be provided in an editable format:
one version identifying all the changes that have been made (for instance, in coloured highlight, in bold text, or tracked changes);
a 'clean' version of the new manuscript that incorporates the changes made, but does not highlight them. This version will be used for typesetting.
Please ensure that any equations included in the paper are editable text and not embedded images.

Please ensure that you include an acknowledgements' section before your reference list/bibliography. This should acknowledge anyone who assisted with your work, but does not

qualify as an author per the guidelines at <https://royalsociety.org/journals/ethics-policies/openness/>.

===PREPARING YOUR REVISION IN SCHOLARONE===

-- Ensure that your data access statement meets the requirements at <https://royalsociety.org/journals/authors/author-guidelines/#data>. You should ensure that you cite the dataset in your reference list. If you have deposited data etc in the Dryad repository, please only include the 'For publication' link at this stage. You should remove the 'For review' link.

-- If you have uploaded ESM files, please ensure you follow the guidance at <https://royalsociety.org/journals/authors/author-guidelines/#supplementary-material> to include a suitable title and informative caption. An example of appropriate titling and captioning may be found at https://figshare.com/articles/Table_S2_from_Is_there_a_trade-off_between_peak_performance_and_performance_breadth_across_temperatures_for_aerobic_sc_ope_in_teleost_fishes_/3843624.

Author's Response to Decision Letter for (RSOS-201874.R0)

See Appendix A.

Decision letter (RSOS-201874.R1)

Dear Dr Silva,

It is a pleasure to accept your manuscript entitled "POLITICAL ARCHITECTURES IN THE MUNICIPALITY OF VARRE-SAI (Brazil): FOR SUSTAINABILITIES IN "SPECIALTY COFFEES" PRODUCTION MANAGEMENT" in its current form for publication in Royal Society Open Science.

on behalf of Dr Agnieszka Latawiec (Associate Editor)
openscience@royalsociety.org

Appendix A

Replies to reviewers' comments:

Reviewers 1 and 2:

- ✓ We appreciate the contribution to a new title, but we believe that the chosen title encompasses more the discussion of the manuscript than the proposed title;
- ✓ The repetitions of expressions and terms are corrected and marked in blue in the correction manuscript;
- ✓ The eleven references not mentioned in the bibliography were placed in the version marked in blue and in the final version of the manuscript;
- ✓ In relation to the reference that an excerpt of the manuscript would refer to changes occurred in the highlighted region, we believe that there was some disagreement about the English version; we actually said that there would be challenges for the sector to be faced by the region and not that we were referring to the actual changes that would have occurred;
- ✓ The 2017 version of the Census was consulted on the IBGE website so that data on coffee production could be updated.

Sincerely
the authors
February 21, 2021